# Prevalence of the Frank's sign by aetiopathogenic stroke subtype: A prospective analysis

Laura Sánchez-Cirera[1‡], Saima Bashir[2‡], Adina Ciscar[3], Carla Marco[2], Verónica Cruz[4], Mikel Terceño[2], Yolanda Silva[2], Joaquín Serena[5]*

1 Fellow of Neurology, Stroke Unit, Department of Neurology, Hospital Universitary Doctor Josep Trueta de Girona, Girona, Spain, 2 Neurologist, Stroke Unit, Department of Neurology, Hospital Universitary Doctor Josep Trueta de Girona, Institut d'Investigació Biomèdica de Girona Dr. Josep Trueta (IDIBGI), Girona, Spain, 3 Student of Medicine, Medicine Faculty, University of Girona, Girona, Spain, 4 Nurse of Neurology, Stroke Unit, Department of Neurology, Hospital Universitary Doctor Josep Trueta de Girona, Girona, Spain, 5 Stroke Unit, Department of Neurology, Hospital Universitary Doctor Josep Trueta de Girona, Institut d'Investigació Biomèdica de Girona Dr. Josep Trueta (IDIBGI), Girona, Spain

‡ These authors contributed equally as first authors to this work.
* jserena.girona.ics@gencat.cat

**Data Availability Statement:** All relevant data are within the manuscript and its Supporting Information files.

## Abstract

### Background and purpose

The Frank's sign is a diagonal earlobe crease running from the tragus to the edge of the auricle at an angle of 45˚. Many studies have associated this sign with coronary artery disease and some with cerebrovascular disease. The objective of this study was to analyse the prevalence of the Frank's sign in patients suffering from acute stroke with a particular focus on its prevalence in each of the five aetiopathogenic stroke subtypes. Special interest is given to embolic stroke of undetermined source (ESUS), correlating the sign with clinical and radiological markers that support an underlying causal profile in this subgroup.

### Methods

Cross-sectional descriptive study including 124 patients admitted consecutively to a stroke unit after suffering an acute stroke. The Frank's sign was evaluated by the same blinded member of the research team from photographs taken of the patients. The stroke subtype was classified following SSS-TOAST criteria and the aetiological study was performed following the ESO guidelines.

### Results

The Frank's sign was present in 75 patients and was more prevalent in patients with an ischaemic stroke in comparison with haemorrhagic stroke (63.9 vs. 37.5, p<0.05). A similar prevalence was found in the different ischaemic stroke subtypes. The Frank's sign was significantly associated with age, particularly in patients older than 70 who had vascular risk factors. Atherosclerotic plaques found in carotid ultrasonography were significantly more frequent in patients with the Frank's sign (63.6%, p<0.05). Analysing the ESUS, we also

**Funding:** JS: 3 -Spanish Ministry of Economy and Competitiveness for grants RETICS-INVICTUS-PLUS (RD0016/0019/0003) funded by Instituto de Salud Carlos III and cofunded by the European Regional Development Fund [ERDF]. -Instituto de Salud Carlos III with a Grant for Health Research (PI16/01540) -Government of Catalonia-Agència de Gestió d'Ajuts Universitaris i de Recerca (2017 SGR 1730). The funders had no role in study design, data collection and analysis, decision to publish, or preparation of the manuscript.

**Competing interests:** The authors have declared that no competing interests exist.

found an association with age and a higher prevalence of the Frank's sign in patients with vascular risk factors and a tendency to a high prevalence of atherosclerosis markers.

## Conclusion

The Frank's sign is prevalent in all aetiopathogenic ischaemic stroke subtypes, including ESUS, where it could be helpful in suspecting the underlying cardioembolic or atherothrombotic origin and guiding the investigation of atherosclerosis in patients with ESUS and the Frank's sign.

## Introduction

Although there is no consensus as to the definition of the Frank's sign and different authors have used variations with regaards to the extent of ear crease, the Frank's sign, or earlobe crease (ELC), is widely defined as a diagonal crease in the earlobe extending from the tragus across the lobule to the rear edge of the auricle at an angle of 45˚ with varying depths.

Although historically the sign can be seen in busts of the Roman emperor Hadrian, who is presumed to have died from cardiac disease, it was first described by Sanders T. Frank in 1973 who, in doing so, suggested a positive relationship between ELC and coronary artery disease (CAD) [1]. Several studies have considered a possible link between ELC and CAD, and many studies have shown that ELC can serve as a marker of atherosclerotic disease and as a sign of an elevated risk of coronary heart disease in asymptomatic individuals [2–7]. This sign has also been associated with HLA-B27, C3-F arteriosclerosis gene and chromosome 11 [8]. Although different studies have tried to identify the pathophysiology of the Frank's sign, the mechanisms underlying the association between ELC and vascular disease remain unclear. In the early 1970s and the 1980s, it was suggested that ELC might be a result of local poor supply from arteries to the earlobe [9]. In a case report, a link is suggested between macrophage activity (which is involved in atherosclerosis), ageing and maintaining earlobe collagen [10], and it has been argued that the ELC and CAD are related to the loss of elastin and to the rupture of elastic fibres in patients with ischaemic heart disease [11]. In a recent histopathological study in 45 consecutive adult patients referred for autopsy, Stoyanov GS et al. analysed samples from both earlobes as well as cardiac samples from all four cardiac compartments. They found a significant correlation between the morphological changes of the myocardium and the presence of the earlobe creases with arterial myoelastofibrosis, Wallerian-like degeneration in peripheral nerves, and deep tissue fibrosis found in the base of the crease [12].

Some studies have described an association between Frank's sign and the presence of diabetes, hypertension [11, 13], myocardial infarction and coronary artery disease [6, 7] in patients of both sexes. According to these studies, this easily identifiable sign could be valuable in the screening of patients at high risk of having silent coronary artery disease.

However, only a few studies have focused on the association between the presence of ELC and cerebrovascular disease [14]. The microvascular damage, the presence of perivascular cellular infiltration and the decreased nitroglycerine-induced vasodilation described in different studies [15] suggest that ELC is associated with endothelial dysfunction, which is known to represent an early stage of systemic atherosclerosis. Additionally, carotid artery intima-media thickness (IMT) and atherosclerotic plaques in the carotid arteries have also been described in association with ELC [16, 17]. These findings suggest that the presence of ELC could serve as a reliable marker of systemic atherosclerosis.

The term ESUS (embolic stroke of undetermined source) describes a type of ischaemic stroke defined as a non-lacunar infarction by neuroimaging, with no internal carotid artery stenosis or dissection of the respective brain supplying artery, and based on the exclusion of other aetiologies such as atrial fibrillation, vasculitis, drug abuse, or coagulopathies [18, 19]. To the best of our knowledge, no studies have analysed the prevalence of ELC stratified by stroke subtype and none has focused on the potential relevance of the presence or absence of ELC in ESUS as a potential marker of the subjacent stroke mechanism in this important and prevalent stroke subtype.

## Material and methods

The study has been submitted and approved by the Ethics Committee of our center (Comitè d'Ètica d'Investigació amb Medicaments CEIm Girona. Approval number 2021.072). This was a cross-sectional descriptive study that enrolled consecutive patients admitted with an acute stroke at the Stroke Unit of our centre from 16[th] September to 16[th] December 2019. Transient ischaemic attacks were included and stroke mimics were excluded. Informed consent was obtained to participate in the study. ELC evaluation was performed by taking photographs of both ears of the patients that were then assessed by a member of the research team, who was blind to the clinical data with a previous informed consent. Patients with potential confounding factors for the evaluation of ELC, such as earrings that had markedly deformed the earlobe, earlobe injuries or tattoos, were excluded from the study.

For the purpose of this study, positive ELC was considered when subjects had a crease or wrinkle extending 45˚ diagonally from the tragus towards the outer border of the earlobe. We considered both the unilateral and bilateral presence of such a crease or wrinkle to be the Frank´s sign (Fig 1).

All patients were submitted to the following examinations: medical history with particular interest in classical stroke risk factors and their treatment, clinical examination, blood tests, 12-lead ECG and continuous Stroke Unit monitoring (blood pressure, heart rate frequency and oxygen saturation), non-contrast CT scan, colour-coded duplex sonography of the supra-aortic trunks, basal colour-coded transcranial ultrasonography, and detection of right-to-left shunt if appropriate. Extensive investigations, including transthoracic echocardiography, were carried out on patients diagnosed as suffering from stroke of uncertain aetiology. Stroke severity was quantified by a certified neurologist using the NIHSS at admission and every 12 hours at the Stroke Unit [20]. CT scan or MRI was carried out at admission. Colour-coded transcranial ultrasonography was carried out systematically and transthoracic echocardiography was performed in all cases of cryptogenic stroke. The suspected cause of stroke was classified as (1) large-artery atherosclerosis, (2) cardioembolism, (3) small-vessel disease, (4) stroke of other determined aetiology, and (5) stroke of undetermined aetiology, in accordance with the Trial of Org 10172 in Acute Stroke Treatment (SSS-TOAST) criteria [21, 22]. The modified Rankin scale (mRS) was measured at hospital discharge and at 3 months, also by a certified neurologist [23].

### Statistical analysis

Results are expressed as frequencies or percentages for categorical variables. Continuous variables are expressed as mean ± standard deviation (SD) and were compared using the Student's t test, or median and quartiles and using the Mann-Whitney tests, depending on whether the distribution was normal or not. We evaluated the normal distribution of the variables using the Kolmogórov-Smirnov test. Proportions between groups were compared using the Chi-square test or t test as appropriate. The importance of ELC in all stroke subtypes and

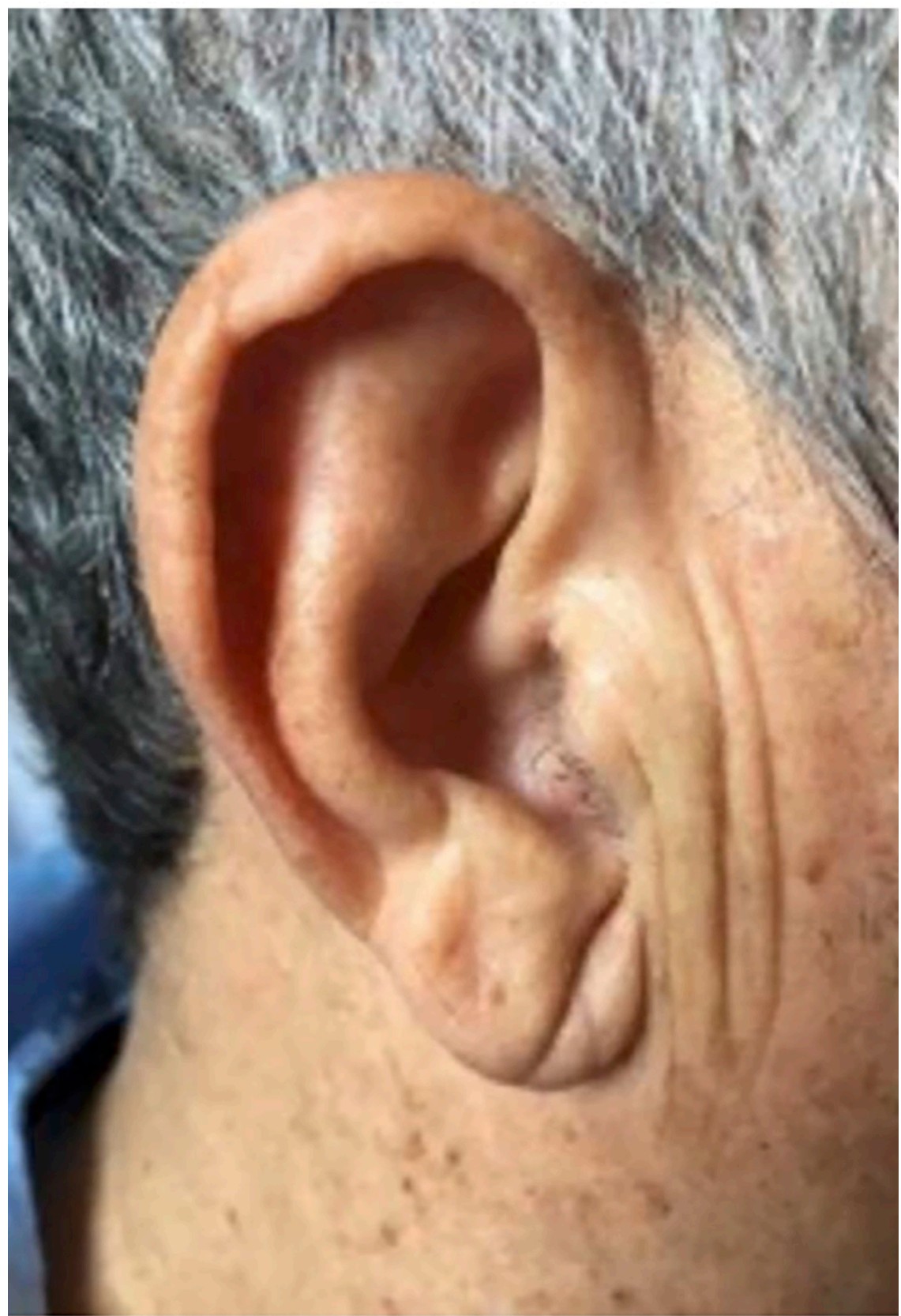

**Fig 1. The Frank's sign.** Diagonal earlobe crease.

specifically in ESUS was assessed by binary logistic regression analysis controlling for independent variables that obtained statistical significance of $p < 0.1$ in the bivariate analysis. Results were expressed as ORs and 95% CIs. For all analyses, significance was taken as a p value of <0.05.

## Results

A total of 124 consecutive patients who were hospitalized with an acute stroke and were eligible to take part in the study were recruited. The mean age of the total population was 70.0 (+/- 14.0), 74.2% were males and 25.8% females. The Frank's sign was present in 75 patients (60.5%). Of the total population, 108 patients (87.1%) had suffered an acute ischaemic stroke (12 of them had a transient ischaemic attack), and 16 (12.9%) a hemorrhagic stroke. There were no relevant differences in the prevalence of ELC by stroke subtypes.

As previously described in the literature [13, 14, 24], we found a strong and significant association between the presence of ELC and age, with ELC being particularly prevalent in patients who were older than 70 years (74.7%). In comparison with ischaemic stroke, we found a lower prevalence of the Frank's sign in haemorrhagic stroke (63.9% vs. 37.5%, p<0.05). The prevalence of ELC had a similar distribution across each aetiopathogenic ischaemic stroke subtype: atherothrombotic (69.2%), cardioembolic (60.0%), lacunar (64.3%) and ESUS (65.2%) (Table 1). As expected, we found a significant association between some classical stroke risk factors, such as hypertension, diabetes, or ex-smokers and ELC, although we observed an inverse correlation in the particular cases of alcoholic habit (>20 g/day) and smokers. On analysing carotid ultrasonographic markers of vascular atherosclerosis, we found a significant relationship between the prevalence of ELC and the presence of non-stenotic atherosclerotic plaques (63.6% vs. 45.2%, p<0.05). Other markers such as stenosis or occlusion of carotid arteries or of other vascular cerebral territories showed an association with the presence of ELC, although these did not reach statistical significance (Table 1). In binary logistic regression analysis, only age and diabetes remain independently associated with ELC (Table 2).

The mRS was measured at hospital discharge in 124 patients and the outcome at 3 months was evaluated in 89 patients. Thirty-five patients failed to attend the appointment scheduled for 3 months after stroke. ELC is associated with a poor outcome (mRS >2) at discharge and at 3 months (76.9% at 3 months) in bivariate analysis (Tables 1 and 3). However, in binary logistic regression analysis only age remains as a predictor of mRankin >2 both at discharge and at 3 months (analysis not shown).

Twenty-three patients (25% of ischaemic strokes) had suffered an ESUS. Among these, we found a high prevalence of patients with ELC (65.2%). In this group, there was also a significant association between ELC and age; whereas only 6 (42.8%) patients under 70 years old presented the Frank's sign, all of the patients over 70 years old (9 patients) had the sign. In this subgroup, ELC is more frequent in patients with anterior circulation strokes (81.8% vs. 18.2%). A higher prevalence of ELC was observed in those patients with arterial hypertension, diabetes, dyslipidemia and obesity in the subgroup of ESUS, although none of these was statistically significant (Fig 2). Similarly to the total population analysis, smoking and alcohol habits seem to be related to the absence of ELC (33.3% and 13.3% respectively), and this was significant in non-smokers (p<0.05).

## Discussion

The association between ELC and CAD has been reported several times since Sanders T. Frank noticed that many patients with ischaemic heart disease had an earlobe crease [3, 6, 7].

**Table 1. Bivariate analysis of the Frank's sign by clinical variables included in the study.**

| | FRANK'S SIGN | | |
|---|---|---|---|
| | **NO** | **YES** | **p value** |
| | **(n = 49)** | **(n = 75)** | |
| Male | 38 (77.6%) | 54 (72.0%) | n.s. |
| Female | 11 (22.4%) | 21 (28.0%) | |
| Age | | | <0.001 |
| <50 | 16 (32.7%) | 5 (6.7%) | |
| 51–59 | 10 (20.4%) | 6 (8.0%) | |
| 60–69 | 12 (24.5%) | 8 (10.7%) | |
| 70–79 | 7 (14.3%) | 29 (38.7%) | |
| >80 | 4 (8.2%) | 27 (36.0%) | |
| Stroke subtype | | | <0.05 |
| Ischaemic | 39 (79.6%) | 69 (92.0%) | |
| Haemorrhagic | 10 (20.4%) | 6 (8.0%) | |
| Ischaemic stroke | | | n.s. |
| Atherothrombotic | 8 (30.8%) | 18 (69.2%) | |
| Cardioembolic | 14 (40.0%) | 21 (60.0%) | |
| Lacunar | 5 (35.7%) | 9 (64.3%) | |
| Cryptogenic (ESUS) | 8 (34.8%) | 15 (65.2%) | |
| Arterial hypertension | 28 (57.1%) | 60 (80.0%) | <0.01 |
| Diabetes mellitus | 9 (18.4%) | 27 (36.0%) | <0.05 |
| Dyslipidemia | 21 (42.9%) | 37 (49.3%) | n.s. |
| Obesity (BMI>30) (n = 79) | 9 (25.7%) | 13 (29.5%) | n.s. |
| Smoker | 22 (44.9%) | 15 (20.0%) | <0.05 |
| Ex-smoker | 10 (20.4%) | 23 (30.7%) | |
| Non-smoker | 17 (34.7%) | 37 (49.3%) | |
| Alcohol intake (>20gr/d) | 18 (36.7%) | 11(14.7%) | <0.05 |
| Ischaemic heart disease | 7 (15.6%) | 8 (11.6%) | n.s |
| Atrial fibrillation | 7 (15.6%) | 16 (23.2%) | n.s |
| Previous ischaemic stroke | 8 (17.4%) | 12 (16.7%) | n.s. |
| Presence of non-stenotic atherosclerotic plaque | 19 (45.2%) | 42 (63.6%) | <0.05 |
| Internal carotid artery stenosis | | | n.s. |
| No stenosis | 34 (79.1%) | 52 (76.5%) | |
| Stenosis >50% | 2 (4.7%) | 7 (10.3%) | |
| Stenosis >70% | 4 (9.3%) | 7 (10.3%) | |
| Occlusion | 3 (7.0%) | 2 (2.9%) | |
| Stenosis other arteries | 11 (25.6%) | 17 (25.0%) | n.s. |
| mRankin ≤ 2 prior to index stroke | 47 (95.9%) | 70 (93.3%) | n.s. |
| mRankin ≤ 2 at discharge | 34 (69.4%) | 38 (50.7%) | <0.05 |
| mRankin ≤ 2 at 3 months | 31 (83.8%) | 32 (61.5%) | <0.05 |

However, the prevalence of ELC in cerebrovascular disease has been less widely studied and there are no systematic studies evaluating its presence across each aetiopathogenic stroke sub-type. The first retrospective preliminary report analysed the prevalence of the Frank's sign in ischaemic stroke patients classified as large-vessel vs. lacunar stroke based on plain CT scans, describing an association between the Frank's sign and non-lacunar ischaemic stroke [25]. Another prospective study that analysed the prevalence of ELC in patients with acute ischae-mic stroke established that ELC could be a predictor of cerebrovascular events [13].

**Table 2. Adjusted odds ratios of ELC for significant clinical variables in bivariate analysis.**

|  | Odds Ratio | 95% CI | p |
|---|---|---|---|
| **Age** | **1.07** | **1.01–1.12** | **0.01** |
| Stroke subtype | 0.51 | 0.04–5.96 | n.s. |
| Arterial hypertension | 2.33 | 0.68–8.10 | n.s. |
| Diabetes mellitus | 5.8 | 1.15–29.49 | <0.05 |
| Smoker | 1.3 | 0.34–4.90 | n.s. |
| Alcohol intake | 0.39 | 0.12–1.29 | n.s. |
| Presence of atherosclerotic plaque | 0.71 | 0.17–2.95 | n.s. |
| mRankin ≤ 2 at discharge | 1.24 | 0.29–5.38 | n.s. |
| mRankin ≤ 2 at 3 months | 2.28 | 0.41–12.70 | n.s. |

Age was included as a continuous variable and so the 7% increased association risk is by every 1 year of age increase. Categorical variables were included as 0 = no or 1 = yes. ELC, or earlobe crease (the Frank's sign).

To the best of our knowledge, this is the first systematic prospective study evaluating the prevalence of the Frank's sign in a consecutive series of acute stroke patients admitted to a stroke unit using an updated aetiopathogenic stroke classification.

In agreement with previous studies, we have found an association between the Frank's sign and age, with it being more prevalent in patients over 70 years. A possible explanation for this is that ELC reflects the ageing processes of skin and arteries [12, 14]. We also found an association of ELC with classical vascular risk factors involved in atherosclerotic disease, especially with hypertension, and diabetes, as well as the presence of non-stenotic carotid plaques. An inverse association with smoking and alcohol intake was found, where non-smokers and patients without alcohol intake presented more ELC. This could be explained by the fact that smokers and patients with alcohol intake of more than 20 gr/day were younger than the global population. In spite of these previously described associations, the logistic regression analysis performed in our series showed that only age and diabetes mellitus remain significantly and independently associated with the presence of ELC. No relevant differences were found between unilateral (25% of patients) and bilateral (75% of patients) Frank's sign.

An interesting finding of our study, not previously referred to in the literature, is that whereas the distribution was similar in atherothrombotic and cardioembolic aetiologies, the prevalence of the Frank's sign was low in haemorrhagic strokes. This may be due to the fact that patients suffering from haemorrhagic strokes are younger, and that the underlying mechanism in this stroke subtype is suspected to be different from that of lacunar or atherothrombotic strokes.

Although ELC has been traditionally associated with atherosclerosis, we found a similar distribution of the Frank's sign in both atherotrombotic and cardioembolic strokes. A possible explanation for this is that the underlying aetiopathogenic mechanism in some cardiac sources of embolism, such as left ventricular akinesia, dilated cardiomyopathy and reduced ejection fraction, is the atherosclerotic disease of coronary arteries [2, 3, 12, 15]. Furthermore, similarly to the prevalence of the Frank's sign, atrial fibrillation is known to be more frequent in older patients. This may be the reason for the higher prevalence than expected of ELC in patients with cardioembolic strokes.

The ELC is prevalent in those patients who have suffered an ESUS. We found that patients who have suffered an ESUS and have the Frank's sign show a more atherosclerotic profile than those without ELC. Patients with an ESUS and with the Frank's sign have a greater prevalence of hypertension, diabetes and dyslipidemia, as well as a greater presence of carotid

**Table 3. Bivariate analysis of modified Rankin Scale at 3 months by baseline clinical variables included in the study.**

| | mRS at 3 months | | |
| --- | --- | --- | --- |
| | mRS ≤ 2 | mRS > 2 | p value |
| | (n = 63) | (n = 26) | |
| Male | 47 (74.6%) | 20 (76.9%) | n.s. |
| Female | 16 (25.4%) | 6 (23.1%) | |
| Age | | | <0.001 |
| <50 | 17 (27.0%) | 0 (0%) | |
| 51–59 | 12 (19.0%) | 1 (3.8%) | |
| 60–69 | 8 (12.7%) | 7 (26.9%) | |
| 70–79 | 19 (30.2%) | 6 (23.1%) | |
| >80 | 7 (11.1%) | 12 (46.2%) | |
| Stroke subtype | | | n.s |
| Ischaemic | 56 (88.9%) | 21 (80.8%) | |
| Haemorrhagic | 7 (11.1%) | 5 (19.2%) | |
| Ischaemic stroke | | | n.s. |
| Atherothrombotic | 12 (23.5%) | 8 (40.0%) | |
| Cardioembolic | 17 (33.3%) | 6 (30.0%) | |
| Lacunar | 10 (19.6%) | 4 (20.0%) | |
| Cryptogenic (ESUS) | 12 (23.5%) | 2 (10.0%) | |
| Frank's sign | 32 (50.8%) | 20 (76.9%) | <0.05 |
| Arterial hypertension | 41 (65.1%) | 22 (84.6%) | n.s |
| Diabetes mellitus | 18 (28.2%) | 8 (30.8%) | n.s |
| Dyslipidemia | 28 (44.4%) | 15 (57.7%) | n.s. |
| Obesity (BMI>30) (n = 79) | 15 (29.4%) | 5 (38.5%) | <0.05 |
| Smoker | 24 (38.1%) | 6 (23.1%) | n.s |
| Ex-smoker | 13 (20.6%) | 10 (38.5%) | |
| Non-smoker | 26 (41.3%) | 10 (38.5%) | |
| Alcohol intake (>20gr/d) | 15 (23.8%) | 8 (30.8%) | n.s |
| Ischaemic heart disease | 8 (12.7%) | 5 (19.2%) | n.s |
| Atrial fibrillation | 9 (14.3%) | 6 (23.1%) | n.s |
| Previous ischaemic stroke | 9 (14.3%) | 8 (30.8%) | n.s. |
| Presence of non-stenotic atherosclerotic plaque | 28 (45.9%) | 18 (85.7%) | <0.05 |
| Internal carotid artery stenosis | | | <0.05 |
| No stenosis | 52 (85.2%) | 12 (60.0%) | |
| Stenosis >50% | 2 (3.3%) | 4 (20.0%) | |
| Stenosis >70% | 4 (6.6%) | 3 (15.0%) | |
| Occlusion | 3 (4.9%) | 1 (5.0%) | |
| Stenosis other arteries | 12 (19.7%) | 8 (40.0%) | n.s. |

atherosclerotic plaques and more stenosis of other intracranial arteries (Fig 2). These results suggest that the presence or absence of ELC in ESUS could be of utility in the search for the underlying cause in ESUS (e.g. high-resolution-MRI techniques looking for aortic or intracranial vulnerable atherosclerotic plaques as an embolic source in ESUS with ELC or prolonged cardiac rhythm monitoring in ESUS without ELC). Further studies are needed focusing on this particular topic.

The main limitation of our study is the small population size, particularly in the ESUS subtype, although it is one of the largest prospective series focused on consecutive stroke patients, using a validated classification. An additional potential limitation is that the assessment of the

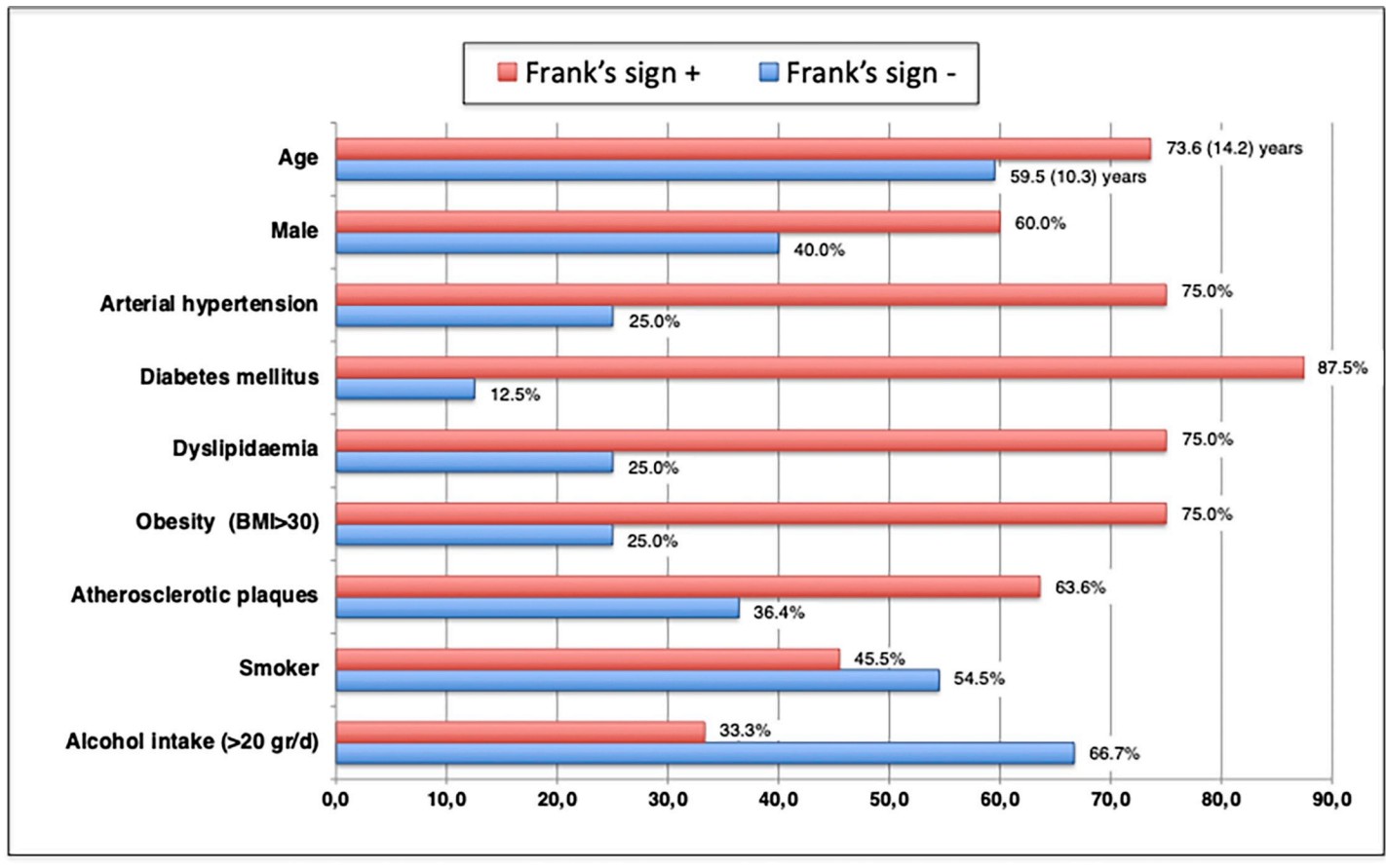

**Fig 2. Frank's sign by clinical variables in ESUS.** Age and prevalence of the main atherosclerotic risk factors in ESUS patients by the presence or not of ELC. Except in the case of age, the X axis represents the percentage of patients with the risk factor recorded on the Y axis. Age is expressed as mean (SD).

Frank's sign was carried out through photographs. In fact, an initial appreciation of ELC was made by a vascular neurologist during the visit to the Stroke Unit, usually on the day of admission and photographs of both ears were taken. After this, a neurologist who was blinded to the neurological symptoms and aetiology of the patient made the evaluation of ELC based exclusively on the photos, which had the additional advantages of permitting the evaluation of the inter-intra-observer variability and of being able to discuss any difficult cases.

## Supporting information

**S1 Data.**
(SAV)

## Author Contributions

**Conceptualization:** Laura Sánchez-Cirera, Saima Bashir, Joaquín Serena.

**Data curation:** Laura Sánchez-Cirera, Saima Bashir, Adina Ciscar, Carla Marco, Verónica Cruz, Joaquín Serena.

**Formal analysis:** Laura Sánchez-Cirera, Saima Bashir, Joaquín Serena.

**Methodology:** Laura Sánchez-Cirera, Saima Bashir, Joaquín Serena.

**Supervision:** Joaquín Serena.

**Visualization:** Mikel Terceño, Yolanda Silva.

**Writing – original draft:** Laura Sánchez-Cirera, Saima Bashir, Joaquín Serena.

**Writing – review & editing:** Laura Sánchez-Cirera, Saima Bashir, Joaquín Serena.

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
