## [Decision Letter · Decision Letter 0]

9 Aug 2021

PONE-D-21-20627

Prevalence of the Frank’s sign by aetiopathogenic stroke subtype: a prospective analysis.

PLOS ONE

Dear Dr. Serena,

Thank you for submitting your manuscript to PLOS ONE. After careful consideration, we feel that it has merit but does not fully meet PLOS ONE’s publication criteria as it currently stands. Therefore, we invite you to submit a revised version of the manuscript that addresses the points raised during the review process.

We look forward to receiving your revised manuscript.

Kind regards,

Juan Manuel Marquez-Romero, M.D., M.Sc.

Academic Editor

PLOS ONE

4. PLOS requires an ORCID iD for the corresponding author in Editorial Manager on papers submitted after December 6th, 2016. Please ensure that you have an ORCID iD and that it is validated in Editorial Manager. To do this, go to ‘Update my Information’ (in the upper left-hand corner of the main menu), and click on the Fetch/Validate link next to the ORCID field. This will take you to the ORCID site and allow you to create a new iD or authenticate a pre-existing iD in Editorial Manager. Please see the following video for instructions on linking an ORCID iD to your Editorial Manager account: https://www.youtube.com/watch?v=_xcclfuvtxQ.

**Comments to the Author**

1. Is the manuscript technically sound, and do the data support the conclusions?

Reviewer #1: No

Reviewer #2: Yes

2. Has the statistical analysis been performed appropriately and rigorously? 

Reviewer #1: No

Reviewer #2: Yes

3. Have the authors made all data underlying the findings in their manuscript fully available?

Reviewer #1: Yes

Reviewer #2: No

4. Is the manuscript presented in an intelligible fashion and written in standard English?

Reviewer #1: Yes

Reviewer #2: Yes

5. Review Comments to the Author

Reviewer #1: It was an honour to review your manuscript. Unfortunately, it requires many corrections, including major changes in data analysis and interpretation. All my comments and explanations were provided in the attached file "Comments".

Reviewer #2: Dear Dr Joaquin Serena:

Thank you for allowing us to read your interesting manuscript. This is a cross-sectional descriptive study that analyzed the prevalence of the Frank’s sign in patients suffering from acute stroke with focus on its prevalence in each aetiopathogenic stroke subtypes. This study concluded that Frank's sign was significantly associated with age, particularly in patients older than 70 with vascular risk factors and that is prevalent in all aetiopathogenic ischemic stroke subtypes, including ESUS. It is a great effort, congratulations. Please consider the following comments.

1.- It should be important to mention in the discussion whether the presence of the unilateral or bilateral Frank's Sign has the same clinical translation according to previous reports, since in the current study both were taken without distinction

.

2.- In the paragraph: As previously described, we found a strong and significant association between the presence of ELC and age, with ELC being particularly prevalent in patients who were older than 70 years (74.7%). Does the term “as previously” mean in previous studies? Because it is the first time that this idea was mentioned in the results of the body of the text.

3.- In the paragraph: On analysing carotid ultrasonographic markers of vascular atherosclerosis, we found a significant relationship between the prevalence of ELC and the presence of atherosclerotic plaques (63.6% vs. 45.2%, p<0.05). Are you referring to non-stenotic plaques? It should be important to clarify.

4.- In the limitations of the study, it should be important to included that the assessment of Frank's sign was carried out through photographs.

5.- It should be important to improve the quality of the tables, especially Table 2, which has disordered information ("Internal carotid artery stenosis")

6.- I think that the conclusion could be improved if the relevance of investigating atherosclerosis in patients with ESUS and Frank's sign is pointed out.

Thanks,

6. PLOS authors have the option to publish the peer review history of their article (what does this mean?). If published, this will include your full peer review and any attached files.

Reviewer #1: No

Reviewer #2: **Yes: **Vanessa Cano-Nigenda

---

## [Author Response · Author response to Decision Letter 0]

27 Sep 2021

Dear Prof 

We are grateful for your comments and suggestions that will clearly improve the final version of the manuscript. 

A point-by-point response to the comments is attached. We would be pleased to provide additional information if requested and to consider further modification of the text if it should be thought necessary.

---

## [Decision Letter · Decision Letter 1]

24 Nov 2021

Prevalence of the Frank’s sign by aetiopathogenic stroke subtype: a prospective analysis.

PONE-D-21-20627R1

Dear Dr. Serena,

We’re pleased to inform you that your manuscript has been judged scientifically suitable for publication and will be formally accepted for publication once it meets all outstanding technical requirements.

Kind regards,

Juan Manuel Marquez-Romero, M.D., M.Sc.

Academic Editor

PLOS ONE

---

## [Editor Report · Acceptance letter]

6 Dec 2021

PONE-D-21-20627R1 

Prevalence of the Frank’s sign by aetiopathogenic stroke subtype: a prospective analysis 

Dear Dr. Serena:

I'm pleased to inform you that your manuscript has been deemed suitable for publication in PLOS ONE. Congratulations! Your manuscript is now with our production department. 

Kind regards, 

on behalf of

Dr. Juan Manuel Marquez-Romero 

Academic Editor

PLOS ONE